# Optimization of Chitosan Properties with the Aim of a Water Resistant Adhesive Development

**DOI:** 10.3390/polym13224031

**Published:** 2021-11-21

**Authors:** Jeanne Silvestre, Cédric Delattre, Philippe Michaud, Hélène de Baynast

**Affiliations:** 1Université Clermont Auvergne, CNRS, Clermont Auvergne INP, Institut Pascal, 63000 Clermont-Ferrand, France; jeanne.silvestre@uca.fr (J.S.); cedric.delattre@uca.fr (C.D.); philippe.michaud@uca.fr (P.M.); 2Institut Universitaire de France (IUF), 1 Rue Descartes, 75005 Paris, France

**Keywords:** chitosan, adhesive, polysaccharide, water-resistance

## Abstract

Chitosan is a bio-sourced polysaccharide widely used in different fields from health to wastewater treatment through food supplements. Another important use of this polymer is adhesion. Indeed, the current demand to replace non-natural and hazardous polymers by greener ones is well present in the adhesive field and open good opportunities for chitosan and its derivatives. However, chitosan is water soluble and exhibits a poor water-resistance in the field of adhesion which reduces the possibilities of its utilization within the paste field. This review focuses on exploration of different ways available to modify the chitosan and transform it into a water-resistant adhesive. The first part concerns the chitosan itself and gives important information from the discovery of chitin to the pure chitosan ready to use. The second part reviews the background information relative to adhesion theories, ideal properties of adhesives and the characteristics of chitosan as an adhesive. The last part focuses on exploration of the possible modification of chitosan to make it a water-resistant chemical adhesive.

## 1. Introduction

Most of the synthetic wood adhesives such as poly(vinyl acetate), epoxy, polyurethane or phenol-formaldehyde adhesives are derived from depleting and non-renewable petrochemical resources [1]. Most of them contain residual toxic chemicals, such as epichlorohydrin, formaldehyde, toluene, or volatile organic compounds (VOCs) [2]. These chemicals are harmful for living beings but also real environmental pollutants. These adhesives present several technical advantages such as their high bond strength or water resistance. However, the growing health and environment concerns encourage wood industries to develop adhesives from bioresources. The bio-based polymers are expected to keep as good bonding properties as the synthetic ones, without the health and environment footprints and without cost increase [3]. Molecules such as tannins, proteins, lignins, or polysaccharides seem to be good candidates for becoming adhesives with the necessary physical, chemical, and mechanical properties. Nevertheless, the high hydrophily of some of them (proteins and polysaccharides) is an obstacle to their development. The most studied polysaccharides as bioadhesives have been the starch, pullulan, levan, dextran, gum arabic, and chitosan [1,4,5]. Among them, the most attractive appeared to be chitosan [3].

Chitosan is a heteropolymer extracted from chitin. This last is found in quantities in a lot of invertebrates such as crustaceans (exoskeleton), insects (cuticles), or fungi [6,7,8,9,10]. Chitosan is produced by alkali deacetylation of chitin. Chitin is the second most abundant natural organic polymer on Earth, after cellulose [11]. Chitin and chitosan are probably among the three more published polysaccharides in the scientific literature along with cellulose and starch [12]. Indeed, when the term « chitosan » is researched in web site of the European Patent Office, more than 190,000 patents are found. However, despite of the availability and the scientific interest for chitin and chitosan, they are not as present on the market as cellulose. This phenomenon could be explained by the structure of chitin which has some specificity compared to cellulose. Indeed, these two natural polysaccharides have a similar arrangement pattern. Both are high molecular weight polymers of glucose linked by β-(1,4) glycosidic linkages [12]. However, in the case of chitin, on the C-2 of glucose units, acetamido groups replace hydroxyl ones leading to a polymer of N-acetyl glucosamines, more hydrophobic than cellulose. The oxygen of this acetamido group is able to form hydrogen bonds with adjacent amino or hydroxyl functions reinforcing the insolubility of chitin. So, the degree of deacetylation, which correspond to the ratio of 2-amino-2-deoxy-d-glucopyranose to 2-acetamido-2-deoxy-d-glucopyranose structural units is directly linked to the chitin insolubility and limits its swelling possibilities in water compared to cellulose [13]. For a degree of deacetylation lower than 0.5, the molecule is considered as chitin whereas for a degree of deacetylation higher than 0.5, it is considered as chitosan [12]. The more this parameter is high, the more the molecule is soluble at acidic pH. Indeed, the pKa of the amino group is about to 6.5. Thus, a protonation of the chitosan occurs at low pH which leads to the solubilization of the product [14,15].

Today, the availability of chitosan on the market is increasing as well as its development in the material fields such as binders, films, or adhesives [2,13,16,17]. Chitosan is a pretty expensive polysaccharide: it is about 10 USD/kg. However, it is usually used in small amount: less than 10% in aqueous solution. This brings the global price to less than 1 USD/L, which is acceptable. The solutions are prepared in 1 to 2% of acetic acid in water and leads to viscous solutions. These are then converted to solid adhesive bonds by the loss of water and solvent or by a chemical setting mechanism. Loss of water takes place either by evaporation or by absorption from porous adherend materials [3]. However, the dry chitosan adhesive is still water sensitive and loose a part of its bonding properties in humid atmosphere. Thus, the aim of this review is to index the different possibilities to make the chitosan an efficient water-resistant adhesive.

## 2. Chitosan

### 2.1. History

Chitin is the second most abundant polysaccharide in nature just after cellulose [18]. It was estimated that around 10^10^–10^11^ tons of this biopolymer is naturally produced each year [19]. Chitin is commonly found in insect cuticles, shells of crustaceans and endoskeleton of mollusks (cephalopods) but also in some mushrooms or yeast envelopes [20]. The discovery of chitin from fungi was firstly attributed to the French Professor Henri Braconnot in 1811 who called it “fongine” [21].

The term “chitin” was used first by Odier in 1823 to design a material extracted from insects (cockchafer elytra) using potassium hydroxide [22]. The word chitin is derived from the French word “chitine” which comes from the Greek “khitōn” meaning covering. The first structural investigation of this macromolecules was done by Lassaigne in 1843 who revealed its nitrogenous nature [23] and was continued with the identification of “glykosamin” in 1876 after hydrolysis of arthropod chitin by Ledderhose [24]. Then, chitin may be considered as the first natural polysaccharide identified and characterized, before the cellulose. The full structure of chitin was finally elucidated in 1946 by Purchase [25].

The history of chitosan had to wait until 1859 to start with the work of Rouget who obtained this biopolymer after hot alkaline treatment of chitin [26].This derivative of chitin was called “chitosan” only from 1894 by Hoppe-Seyler [27]. It was not until the 2000s that the scientific community and industries began to investigate the potential of chitin but above all of chitosan in several fields of applications (materials, nutraceuticals, antimicrobial agent, etc.) as a non-toxic, environmentally friendly and biodegradable polysaccharide with inherent antimicrobial activity.

### 2.2. Structure

Chitin and chitosan are copolymers of β-(1,4) linked N-acetyl-d-glucosamine and d-glucosamine making them analogous of cellulose (β-(1,4) linked d-glucoses). β-(1,4) linkages in the backbone of chitin and chitosan are formed between the hydroxyl of the anomeric carbon of one residue and the secondary hydroxyl of the C4 of another residue leading to a repeating unit. Unlike cellulose, chitosan contains up to 5–8% of nitrogen. As glycosidic unit are linked by either linkage between C-1 of one glycosidic unit to C-4 of an adjacent one, distinct “left” and “right” directions can be assigned to each polymer designating three crystal types of chitosans: α, β, and γ.

The degree of deacetylation (DD) of this biopolymer defines its belonging to chitin or chitosan. Even if the cut-off between these two kinds of polysaccharides has no consensus, the frontier seems to be between 40 and 75% [28]. In the commercial arena, the term chitosan is used for deacetylated chitin with a DD above 60% [12]. Some authors reported that when the percentage of N-acetyl glucosamine is lower than glucosamine, the biopolymer is called chitosan [29]. Depending on their chitin source and extraction processes, molecular weight of chitosan is between 10 and 1000 kDa [28,30].

Primary and secondary hydroxyl groups and amine group are present in each repeating molecule of chitosan. Amine groups are suitable for various typical reactions of amines [31]. Chitosan being less crystalline than chitin, it is more accessible to reagents. Chitosan, contrary to chitin, is soluble in acidic aqueous solutions due to its polycationic nature at acidic pH. This solubility is due to the protonation of amine groups which gives chitosan a polyelectrolytic nature. The functional properties of chitosan are strongly dependent on DD, molecular weight, and their sources.

### 2.3. Production

The main sources of commercial chitosan are fungi, crustaceans, and squids even if recent developments intend to exploit chitin from insects as a new source of chitosan [29,32]. Indeed exoskeleton of insects is an abundant by-products of protein production for food using this raw material [33]. The main source of commercial crustacean production is waste from the seafood industry (crab and shrimp shell wastes). The world market of chitosan, dominated by water treatment application, was evaluated at USD 476.6 million in 2016, and is expected to reach USD 1088.0 million by 2022 [34].

Grounded crab and shrimp shell wastes are firstly treated with diluted hydrochloric acid to remove calcium carbonates (Figure 1). Proteins are then eliminated after their solubilization at alkaline pH. By the way, a part of acetyl groups is hydrolyzed. The solubilized proteins could be collected at this step after precipitation at pH 4. They are recycled locally as fertilizer or for animal feeds notably in India and China. Chitosan is finally obtained after alkaline treatment with NaOH at high concentrations (between 40 and 45% (*w*/*w*)) during 1–3 h at 120 °C under nitrogen [35]. Time, NaOH concentration, and temperature affect the final DD.

This process is highly pollutant and produces large quantities of wastewaters. They are generally treated on-site before to be discharged in environment. Indeed, the sole production of 1 ton of chitosan from grounded crab and shrimp shell required 6300 kg of HCl and 1800 kg of NaOH. So, the impact of chitosan production on environment is strong. The first life cycle analysis of chitosan production was done in 2018 considering manufactures of chitosan in India and Europe [36]. Results showed that the European and Indian chitosans have different environmental profiles reflecting their different supply chains (raw materials), production locations and applications.

Chitin and then chitosan from various sources are commercially produced by some major players in many countries including Germany (Heppe Medical Chitosan GmbH, Saale, Germany), Belgium (Kitozyme S.A., Herstal, Belgium), United States (Agratech International Inc., Denville, New Jersey, USA), Norway (Advanced Biopolymers AS, Trondheim, Norway; NovaMatrix, Sandvika, Norway), China (G.T.C. Bio Corporation Qingdao, Shandong, China; Golden-Shell Pharmaceuticals Co. Ltd., Yuhuan Taizhou, China; Qingdao Yunzhou Biochemsitry Co. Ltd., Qingdao, China), Iceland (Primex ehf, Siglufjordur, Iceland), and India is the leader (Mahtani Chitosan Pvt. Ltd, Gujarat, India; Panvo Organic Private Co. Ltd., Tamil Nadu, India). The size of world chitosan market is confidential. A recent report valued the chitosan the chitosan market size at 1.7 billion USD in 2019 with a projection to reach 4.7 billion in 2027 (-https://www.alliedmarketresearch.com/chitosan-market, 20 November 21). The main applications are currently the use of this polysaccharide as fat blocker and for water treatments as a flocculant. This last application appears as the most lucrative segment.

## 3. Adhesion Principles

### 3.1. Adhesion Theories

The adhesion is very complex because it involves rheology, thermodynamic, chemical, and physical links and surface states. The mechanisms of adhesion can be grouped into three types [37]: -Mechanical adhesion which occurs thanks to mechanical interlocking,-Specific adhesion which concerns the bonding between the adhesive and the adherence,-Effective adhesion: a combination between the two others.

The mechanical adhesion theory is the oldest, developed by Mc Bain and Hopkins in 1925 [38]. Mechanical interlocking of the adhesive occurs on the asperities present at the surface of the adherent and leads to a mechanical bond between the two surfaces. The roughness and the porosity of the surfaces are favorable to good adhesion provided that adhesive has a good wettability. The influence of the wettability (specific adhesion) has been developed in the thermodynamic theory of adhesion by Sharpe and Schonhorn [39]. At thermodynamic equilibrium, a drop of liquid adhesive has a contact angle with the adherend substrate. This equilibrium satisfies the Young-Dupré equation (Equation (1)).
(1)$$\gamma_{S}=\gamma_{SL}+\gamma_{L}\cos\theta$$
where *γ**_S_* is the surface free energy of the solid substrate; *γ**_L_* is the surface free energy of the adhesive drop; *γ_SL_* is the interfacial free energy between the adherend substrate and the adhesive drop; and *θ* is the contact angle between adherend and adhesive liquid interfaces. If the contact angle *θ* is inferior to π/2, adhesion forces are superior to cohesive forces and the liquid can spread. We can note that the contact angle *θ* is disturbed by the roughness of the adherend. Dupré and Dupré have shown that the surface free energies are associated to the work required to separate a unit area of solid–liquid interface [40]. The work is related to the existing intermolecular forces between adherend and adhesive. Intermolecular forces can be Van der Waals forces or hydrogen bonds. 

Among the interactions between adhesive and adherend, the electrostatic attraction theory is based on the different electronic bond structures. Electrostatic forces are formed at the interface between the adhesive and the adherend and adhesive-adherend system can be considered as a plan condenser [41]. Fowkes [42] has shown that the implementation of this system depends on the formation of hydrogen bonds between a hydrogen donor and a hydrogen acceptor. This model can be applied in the case of metallic surface [43] but not with wood for example [37].

Another model of adhesion was proposed by Voytskill based on the diffusion or the interdiffusion at interfaces [44]. It can be applied only if adhesive and adherend are mutually miscible and compatible, i.e., between two polymers. However, Voystskill et al. have shown that the theory can also be used in wood application if temperature implementation is high during a long pressing time [44]. In this case, lignin plays an important role as a thermoplastic polymer [45].

The phenomenon of adhesion is complex and simultaneously appeals to several of these theories. The adhesion must not stop at the interface phenomena but must include what happens in the two related materials in an interface. This is what has shown Marra in the case of wood adhesion [46,47]. He proposed that the interface can been seen as a succession of several layers between pure adhesive phase and wood cells: adhesive film/intra-adhesive boundary layers/adhesive-adherend interface/adherend surface and adherend proper [45]. Marra introduced the notion of adhesive boundary layer.

Adhesive is used at liquid state; this implies that adhesive bond formation is a dynamic process with macroscopic and microscopic scales. Acetic acid is generally used to solubilize chitosan in order to optimize its mechanical properties and its adhesives properties [48]. It has been shown that interaction between chitosan and acid solution influence spatial configuration of the chitosan molecules during solidification. In comparison with lactic, malic and citric acids, acetic acid leads to better mechanical properties [49]. 

### 3.2. The Chitosan as an Adhesive

Molecular weight (MW) and deacetylation degree modify physico-chemical properties of chitosan and influence its adhesive power. 

Studies on chitosan diluted solutions have shown that molecular weight influenced the conformation of chitosans. Low molecular weight chitosans (<148 kDa) were in a rod shape whereas high molecular weight chitosan (>223 kDa) were in a random coil conformation [50,51]. It can be attributed to differences in intra-molecular hydrogen bonds and/or differences in charge distribution. Bulk chitosan forms interconnected porous structure upon freezing or hydrophilizing [52]. Patel et al. have observed the same behavior with chitosan used as adhesive on aluminum specimen [4]. Lee et al. studied interface interaction in chitosan/chitosan cohesion and chitosan/mica adhesion [53]. They have shown that in the case of chitosan versus chitosan cohesion (symmetric configuration), increasing the contact time leads in higher cohesive energy. This phenomenon is certainly due to molecular rearrangement of chitosan: it changes from a relaxed 2-fold helix to an extended 2-fold helix which is more favorable to hydrogen bonds formation and hydrophobic interaction and thus, more favorable to adhesion.

Qun and Ajun have observed that for a high molecular weight chitosan, surface tension decreases when concentration increases [54]. They attributed this phenomenon to the inter molecular hydrogen bonding formed. They have worked with diluted solutions i.e., with concentration inferior to critical overlap concentration. The critical overlap concentration C* was estimated to 1.45 g·L^−1^ with a chitosan having a DD of 90% and a molecular weight of 98 kDa [55]. It was estimated to 1.05 g·L^−1^ with a chitosan having a DD of 88% and a molecular weight of 193 kDa [56] while it has a value between 2.8 and 3.1 g·L^−1^ for a chitosan having a DD of 85% and an molecular weight of 190 kDa [57,58]. For a concentration of 0.5% *w*/*v* (i.e., >C*), Uragami et al. have shown chitosan had a surface tension of 64 mN·m^−1^ at the initial time which decreased to 41 mN·m^−1^ after a long time [59]. The concentration of acetic acid influences the surface tension. According to Geng et al., for 7% *w*/*v* chitosan (106 kDa), surface tension decreased from 54.6 mN·m^−1^ to 31.5 mN·m^−1^ when acetic acid concentration increased from 10% to 90% [60]. To have a good molecular interaction, the adhesive surface tension has to be lower or equal to the adherent one. As chitosan exhibits a low surface tension, it is lower than most of the adherent ones and favorable to good interactions.

Viscosity of chitosan solution is function of concentration and/or molecular weight and decreases with temperature [61]. For example, chitosan solution of 4% *w*/*w* has a non-Newtonian behavior. For this same concentration, the flux changes according to the molecular weight. For low MW, chitosan shows pseudoplastic flow while for medium MW, chitosan shows plastic flow [62]. Thus, playing with the different parameters mentioned above gives the possibility of having a wide range of solubilized chitosans. 

Chitosan structure also includes several sites conducive to interactions. The amino group permits to obtain a positively charged chitosan in acidic environment. These cations interact with negatively charged adherent such as metals, wood or mica and lead to electrostatic adhesion forces [1,4,53]. Moreover, hydrogen bonds, Van der Waals interactions and covalent bonds between the chitosan and the adherent are made possible thanks to the presence of the amino group and the numerous hydroxyl groups.

Thus, chitosan exhibits plenty of physico-chemicals characteristics suitable for a good adhesive. It has been demonstrated that this polysaccharide shows very good adhesion strength which could be compare with the adhesives available on the market. However, chitosan is hydrophile and does not resist to water [63]. It could be interesting to study how to improve its water resistance properties in order to make it a resilient adhesive.

## 4. The Various Modifications of Chitosan in Order to Optimize Its Adhesion Properties

### 4.1. Crosslinking

To improve mechanical properties and chemical stability of polymers to make water-resistant adhesives, the main strategy is to synthetize cross-linked systems. By cross-linking, molecular entanglement could be enhanced, which results in slower relaxation time, lower relaxation ability of the polymeric chains, and reduces the water accessibility within the network. For chitosan, crosslinking is recognized as a most promising modification because its abundant amino and hydroxyl groups can react with other activated function groups [64]. The products created by chemical or physical cross-linking can be achieved using either synthetic or natural-based agents [65]. It consists of formation of bridges between reactive groups of polymer chains. Depending on the selected crosslinker and operating conditions, the cross-linking mechanism may be ionic or covalent [66]. Indeed, chitosan is only soluble in its polycationic form. Thus, it may interact with anionic or polyanionic molecules, creating ionic complexes. 

#### 4.1.1. Covalent Bond

Covalent cross-linkers, such as formaldehyde or glutaraldehyde, give products with stable chemical and physical parameters by creating strong chemical bonds between the cross-linked chains. However, the presence of unreacted products from crosslinkers or by-products formed throughout the reaction may conduct to a potential toxicity of the final commodity [65]. Thus, the obtained hydrogel is generally washed to get rid of these undesirable products.

Classical chemical organic reactions such as the reaction between amines and carbonyls or amines and epoxides are usually employed to cross-link chitosan. Indeed, reagents as glutaraldehyde [67,68], epichlorhydrin [69,70], and ethylene glycol diglycidyl ether [71,72,73] are common cross-linkers and have been widely used in cross-linking of chitosan [74]. However, these reagents are not very safe for living beings and for the environment.

A current challenge about chitosan cross-linkers is the chemical stabilization thanks to nontoxic reagents which could lead to the modulation of the properties for a prolonged period. Thus, an alternative reagent could be used to create a stable and biocompatible cross-linked product [65]. Citric acid, a natural metabolite, is a non-toxic, renewable and bio-based reagent which could react with the chitosan amines to form an amide structure [1]. It is produced commercially by fermentation of sucrose or glucose or black liquor from wood pulping [75]. As citric acid includes several carboxylic acids, it could lead to the creation of a cross-linked chitosan (Figure 2). The formed links could be intermolecular bonds or intramolecular bonds. The use of acid citric in order to improve the water repellency of chitosan has already been studied and has shown interesting results such as an increase of the water contact angle [75].

One of the most common cross-linker from bioresources is genipin. This crystalline and well defined chemical compound exhibits a reduced cytotoxicity compared to the different cross-linkers mentioned above [76,77,78]. Genipin is obtained from a compound traditionally used in Chinese medicine; the geniposide. This last may be isolated from the fruits of *Gardenia Jaminoides Ellis* by a direct chemical procedure or by a microbiological process [65,78]. Chitosan cross-linking by the genipin is easy to implement as both reagents are water-soluble in acidic environments and the reaction mechanism is well understood for a variety of experimental conditions. This mechanism may occur with different pathways [65,79]. Indeed, under mild acidic or neutral conditions, the reaction between chitosan and genipin involves two different steps (Figure 3). The first one is the nucleophile attack of amino groups of chitosan on the olefinic carbon atom at C3 of genipin, followed by opening the dihydropyran ring and carbonyl group attacked by the secondary amino group newly formed. From here, the second step reaction takes place, and two schemes may occur. According to scheme A, an amide is formed thanks to the nucleophile attack of amino group of chitosan to carboxyl group of genipin. According to scheme B, an oxygen radical-induced polymerization of genipin might be produced between genipin molecules already linked to amino groups of chitosan. This could lead to the cross-linking of chitosan chains by genipin molecules or even by genipin copolymers that have a high conjugation of C–C double bond [79,80]. However, despite genipin exhibiting hopeful characteristics, it is not as widespread as common cross-linkers. Indeed, genipin is far more expensive and this may curb consumers.

Inspired by the quinone-tanning process which occurs in nature [81], chitosan-phenolic chemical cross-linking systems have also to be studied. They would be able to improve the chitosan adhesive strength and may improve its water-resistance [82,83]. These systems are composed with chitosan, a phenolic compound, such as dopamine or methylcatechol and a phenol-oxidative enzyme, such as laccase or tyrosinase. The enzyme is necessary to oxidize the phenolic compound via a single electron oxidation to form phenolic free radicals. These radicals can couple with each other and form dimers or even polymers. The phenolic free radicals would react with oxygen to form quinones following the Michael addition reaction. At high temperature in the presence of oxygen, a phenolic compound can also be directly oxidized in quinones. These quinones could react with amino groups following different cross-linking reactions (Figure 4).

#### 4.1.2. Ionic Interactions 

In terms of toxicity, ionic cross-linkers such as phosphates or sulfates, are relatively safe. In addition, with a physical cross-linking system, the thermosensitive character of the polymer is maintained [84]. However, because of the nature of the formed bounds, that involves weak interactions such as electrostatic, dipole–dipole but also hydrogen and hydrophobic interactions, the chitosan based complexes have poorer mechanical properties than the systems formed with covalent bounds [66] (Figure 5). In addition, the presence of numerous charges to form electrostatic interactions gives to cross-linked chitosan a hydrophilic character.

Chitosan, as a cationic polyelectrolyte, is able to react with negative reagents to form ionic interactions and to create a cross-linked system. Acids such as sulfuric acid is a good candidate for this process [85]. It would protonate chitosan to form the ammonium cations thanks to its acidic properties and bring the cross-linker negative group -SO_4_^2−^. Protonation is the limiting step for the sulfuric acid reaction with chitosan and the prerequisite for ionic cross-linking. Another interesting acid for chitosan ionic cross-linking is citric acid. Based on Gohil methods, its negatively charged carboxylate groups could interfere with the positive charges of chitosan [86].

Another common physical cross-linking agent is the inorganic compounds containing phosphor. One of the most common is tripolyphosphate (TPP) [73] but others such as glycerol-phosphate are also used [84]. Here, the protonation of the chitosan has to be performed upstream, with acid acetic for example. Then, the positively charged chitosan can be treated with the polyanion in order to obtain a cross-linked chitosan system [73].

Polyelectronic complexes can also be formed by reaction between chitosan and various natural and synthetics anionic polyelectrolytes such as sodium alginate [87]. These anionic polyelectrolytes can be other polysaccharides, proteins, DNA or different synthetic anionic polyelectrolytes. The formed complexes are usually hydrogels and generally insoluble in water. A lot of studies have already been carried out due to the wide variety of applications of the complexes formed [66].

#### 4.1.3. Ionic/Covalent Co-Cross-Linking

In order to maintain the thermosensitive character of chitosan associated to an ionic cross-linker and to keep the improved mechanical properties and chemical stability usually linked to a covalent system, it is possible to combine physical and chemical cross-linking and to form a co-cross-linked gel [84,88] (Figure 6). By choosing appropriate co-cross-linkers such as tripolyphosphate (TPP, as a physic cross-linker) and genipin (as a chemical cross-linker), the bonds are pH dependent. At lower pHs, the ionic crosslinking dominates. However, chemical crosslinking was major at neutral and basic pHs [88]. Thus, varying the pH can be made to control the bonds strength and so, the thermosensitive character and mechanical properties of created network.

A lot of reagents are available to form physical or chemical cross-linked systems with better properties than chitosan alone. However, in order to develop a water-resistant chitosan, its hydrophilicity has to be reduced. This parameter could be modified after grafting a hydrophobic reagent on it.

### 4.2. Hydrophobization

Hydrophobic associating water-soluble polymers have been studied for decades now [89]. Hydrophobic substituents are incorporated into polymer molecules through chemical grafting or suitable copolymerization procedures. Alkyl chains are the most common hydrophobic groups, but other substituents can be bond to chitosan to reduce its hydrophilicity.

#### 4.2.1. Hydrophobic Alkyl Chains Grafting

It has been demonstrated that chitosan substituted with alkyl chains containing a minimum of six carbons illustrates hydrophobic interactions in solution [89,90]. Alkyl chains can have different shapes, the most popular ones being linear but they can also be branched or even cyclic or aromatic [91,92]. In order to graft one of these chains to the chitosan, different reactions can be set up.

One of them is the N-alkylation and more precisely, the reductive alkylation of chitosan [93]. This reaction is a versatile and specific method for creating covalent bonds between a carbonyl and an amine. The first step is a slow reaction between the chitosan amino group and an aldehyde or a ketone containing an alkyl chain. For purposes of having a homogenous accessibility of reactive sites, chitosan has to be precipitated by basic neutralization [94]. The obtained product is a cationic Schiff base which is reduced in a second stage to form a saturated bond. The reducing agent can be sodium hydroborate (NaBH_4_) or selephenol (PhSeH) but the better reactivity and selectivity of sodium cyanoborohydride (NaBH_3_CN) make it a more common reducing reagent [89] (Figure 7). The reductive alkylation of chitosan has already been studied with a view to improve its water resistance for bonding applications and interesting results have been demonstrated [95].

Another way to graft an alkyl chain on chitosan is N-acylation. This reaction is a specific amidation between an acid containing an alkyl chain and the amino groups from chitosan [93]. In order to proceed to an easier reaction, acid derivatives such as acyl chlorides and anhydrides are usually used [96]. An acid derivative is able to react with both hydroxyl and amino groups and thus on the two hydroxyl groups and the amino group on the repeating unit of chitosan. The O-acylation leading to an ester linkage between alkyl chains and chitosan gives also hydrophobic characteristics to the biobased polymer and could be interesting to enhance its water-resistance properties. Several authors have shown that it is possible to control the regioselectivity of the acylation reaction to graft the alkyl chain on either amino, hydroxyls, or on both groups [96]. This control can be perform using phthalic anhydrides and trityl groups (Figure 8).

#### 4.2.2. Other Hydrophobic Reagents

Acylation and alkylation can also be adapted to graft other hydrophobic groups on polysaccharides. Indeed, alkyl chains are the most basic hydrophobic reagents, but a lot of acid derivatives can be bond to chitosan to reduce its hydrophilicity. For example, polyfluorinated chemicals (PFCs) have already been grafted to chitosan to improve its water repellency [75]. Although PFCs are harmful for the environment as well as for human and animal health, it is important to notice their unique properties [97]. Indeed, most of them are insoluble in aqueous solution and repel water. In addition, they are extremely stable and show very low surface tensions. As mentioned earlier, the lower the surface tension is, the easier the spreading on the adherent is. It has been proved that it is possible to link perfluorinated acid derivatives by amide bonds onto the chitosan [98]. This reaction leads to the hydrophobicity chitosan control up to a point.

Other ways of synthesis leading to the chitosan hydrophobization can also be cited. One of them is the copolymerization of chitosan by an in situ microwave assisted condensation reaction [99]. Pal et al. have used this method to synthetize a chitosan-graft-lactic acid oligomer [100]. Poly(lactic acid) is a biobased, nontoxic biodegradable compound. The reaction between the lactic acid carboxylic acid and the amino group of the chitosan takes place under microwave radiation and convective heating. The grafted groups act as hydrophobic tails within the copolymer formed and may lead to the water resistance of chitosan.

### 4.3. Formulation

The formulation is a mixing operation involving different compounds which leads to the creation of a stable and homogenous material. It does not request reaction between the chemicals. In the case of the chitosan improvement, the aim is to add an ingredient which will modify the polysaccharide properties and enhance its water resistance.

#### 4.3.1. Plasticizers Adding

Plasticizers are plastic additives commonly used to increase structural flexibility and processability of polymers [101]. These properties modifications are set up by the reduction of the second order transition temperature [102]. Generally, plasticizers are low molecular weight chemicals and take the form of resins or liquids. They might form bonds with the polymer chains, reducing intermolecular forces and space the chains apart. This phenomenon leads to a higher mobility of the polymer which will be softer, more easily deformable, and thus more resistant. Plasticizers locate in the amorphous area of polymers without modifying the structure and size of the crystalline areas [103].

Plasticizers can be external or internal. An external plasticizer is not linked to the polymer by primary bonds and can be easily separated from it (by evaporation or migration for example). Conversely, internal plasticizers are well linked to the polymer and remain part of the product.

The most common used plasticizers are phthalates and phosphates. However, these plasticizers are not the greenest ones. As a result of the growing health and environment concerns and the development of natural and nontoxic biopolymers such as chitosan, compatible plasticizers are favored. Polyols are a very common class of compounds which are used as plasticizers for biodegradable polymers [101]. Glycerol, sorbitol or glucose have been formulated with chitosan to improve its properties [1,4,104,105,106,107]. Mechanical properties of chitosan are enhanced with glycerol or glucose [4,104,105,107] but not with sorbitol. Moreover, none of these plasticizers have demonstrated water resistance improvement of the polysaccharide excepted glycerol which, in a large amount leads to the reduction of the water absorption by chitosan films [75]. This observation could be explained by the fact that all of these plasticizers present hydrophilic properties; their addition to the chitosan leads to the increase of its water affinity [108]. Therefore, hydrophobic plasticizers should be incorporated within the polymer structure. This kind of plasticizers, such as ester citrate derivatives, would enhance the water-barrier properties of chitosan. 

#### 4.3.2. Interpenetrating Polymers

Modifying polymer properties could also be performed via the creation of an interpenetrating polymer network (IPN). IPNs consist of a mixture of at least two cross-linked networks which are mixed or dispersed at a molecular segmental level [109]. If the polymers are not cross-linked together, that is, one of them is cross-linked and the other one is in linear form, the system is called semi-IPN. Otherwise, when both polymers are cross-linked, the term used is full-IPN [110] (Figure 9). 

IPNs could be a way to enhance the chitosan mechanical properties and reduce water accessibility between the polymer chains. This could be set up by using cross-linking in addition to the decline of chitosan hydrophilicity thanks to the incorporation of a second polymer with hydrophobic properties such as polyethylene. 

## 5. Conclusions

Chitosan is a bio-sourced, biodegradable, and nontoxic polymer which has been widely studied over the last years. It is one of the rare naturally occurrent positively charged polysaccharide containing numerous amine and hydroxyl groups within its backbone. The resultant properties have made chitosan a biopolymer used in various fields such as food, biomedical applications, or bonding. Indeed, chitosan has demonstrated very interesting properties to use it as an adhesive. However, its water-resistance is a real curb for this application and has to be enhanced. Thus, chitosan modifications such as cross-linking, hydrophobization, or formulation have to be implemented. This would enhance the mechanical properties and reduce its water affinity to make it a strong water-resistant adhesive, making it competitive with the synthetic adhesives. Of course, these modifications must be as green as possible, in order to maintain the attraction linked to the eco-friendly characteristics of chitosan. In addition, they must not increase chitosan cost significantly in order to keep the competitive price of chitosan as an adhesive.

## Figures and Tables

**Figure 1 polymers-13-04031-f001:**
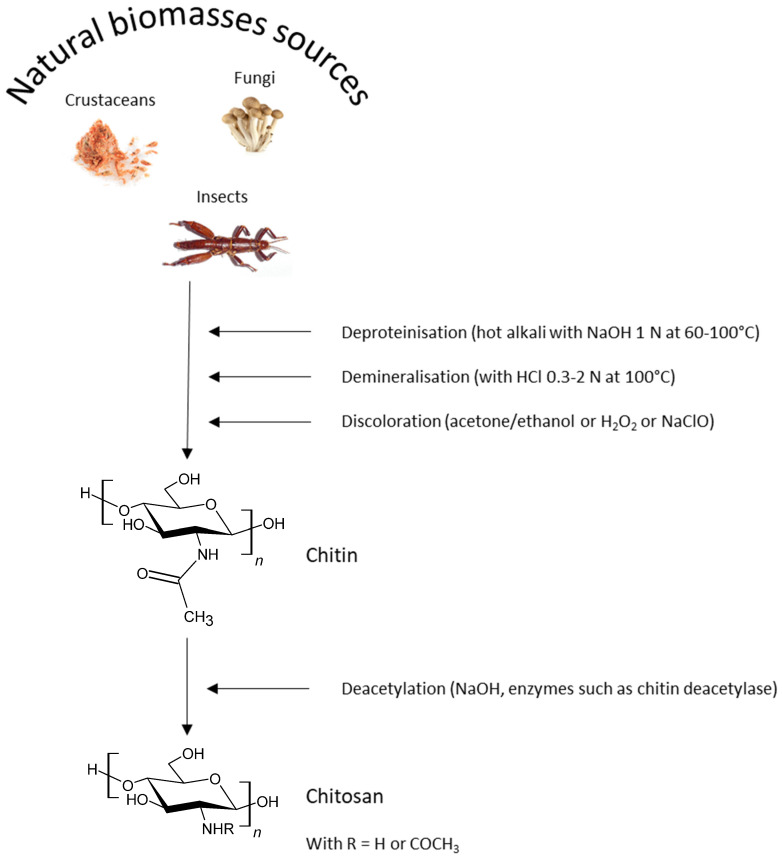
General processes for chitosan production.

**Figure 2 polymers-13-04031-f002:**
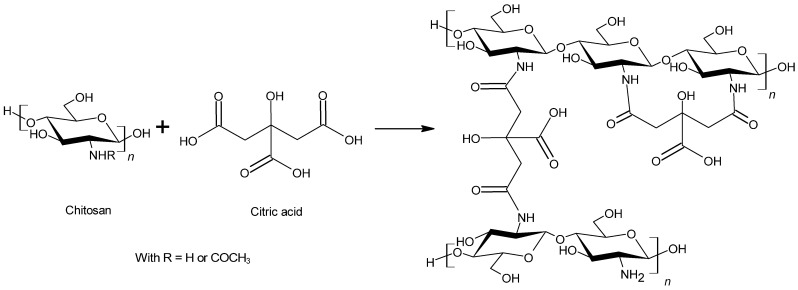
Covalent cross-linking reaction between chitosan and citric acid.

**Figure 3 polymers-13-04031-f003:**
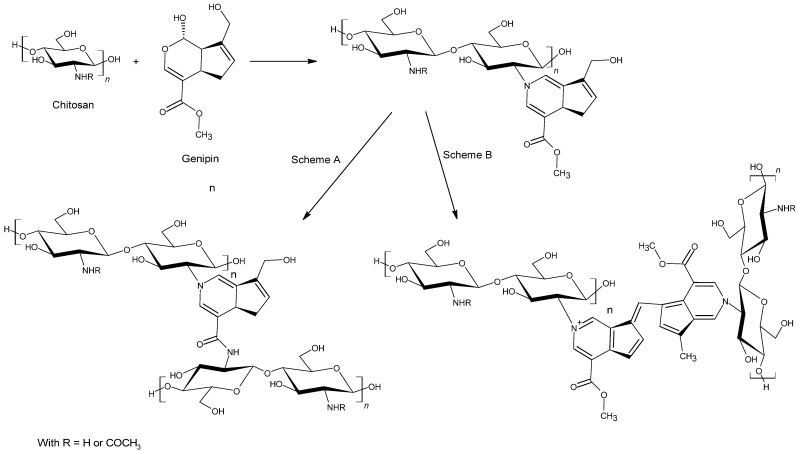
Covalent cross-linking reaction between chitosan and genipin.

**Figure 4 polymers-13-04031-f004:**
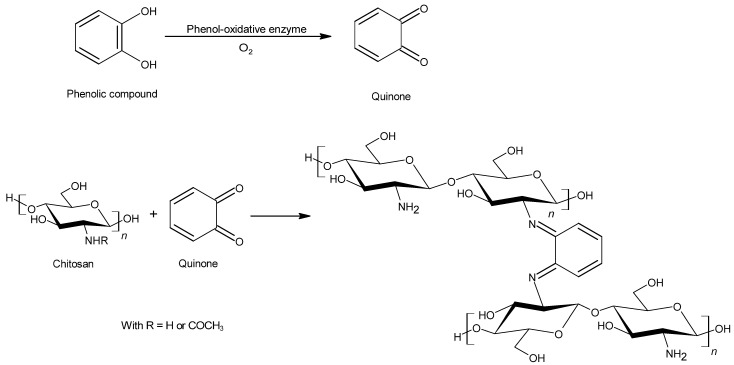
Covalent cross-linking reaction between chitosan and a phenolic compound (pyrocatechol).

**Figure 5 polymers-13-04031-f005:**
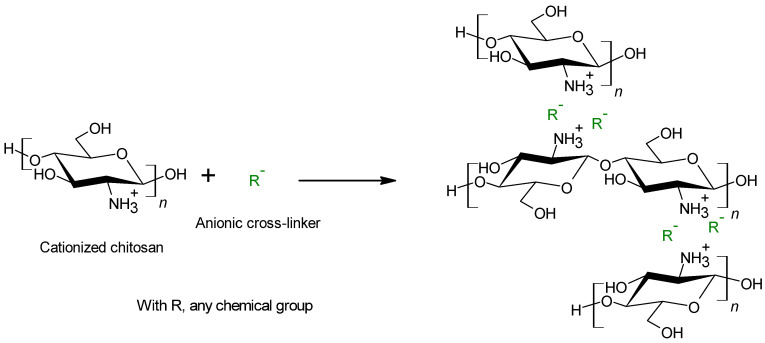
Ionic cross-linking reaction between chitosan and a negatively charged cross-linker.

**Figure 6 polymers-13-04031-f006:**
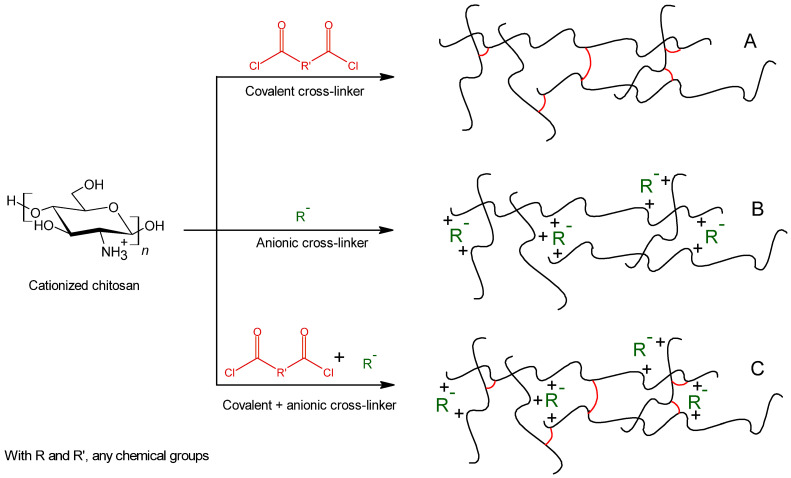
Schematic representation of chitosan covalent cross-linking (**A**), ionic cross-linking (**B**) and co-cross-linking (**C**).

**Figure 7 polymers-13-04031-f007:**
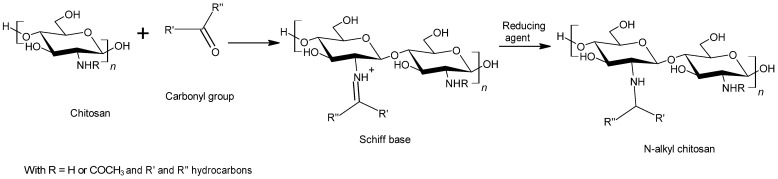
Alkylation reaction process of chitosan.

**Figure 8 polymers-13-04031-f008:**
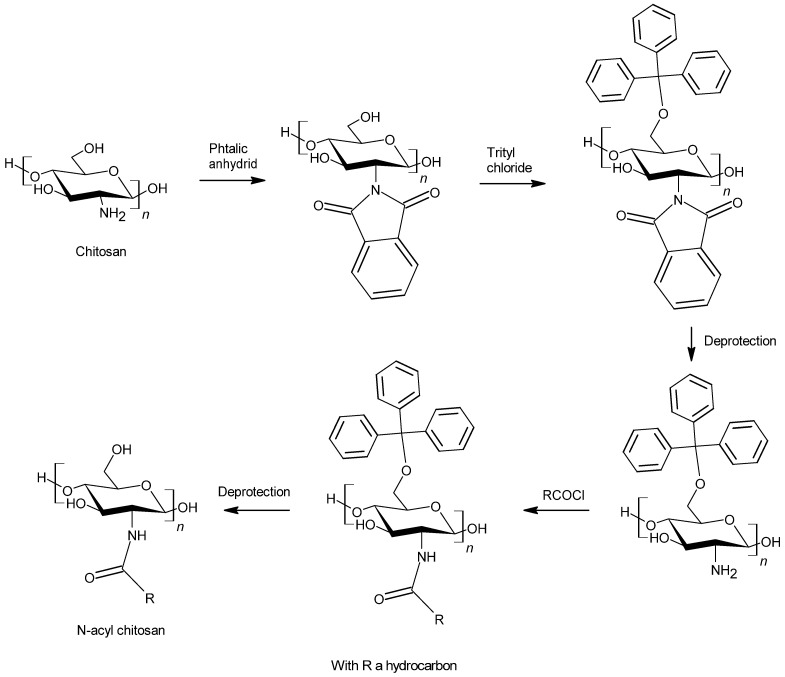
Regioselective example of chitosan acylation.

**Figure 9 polymers-13-04031-f009:**
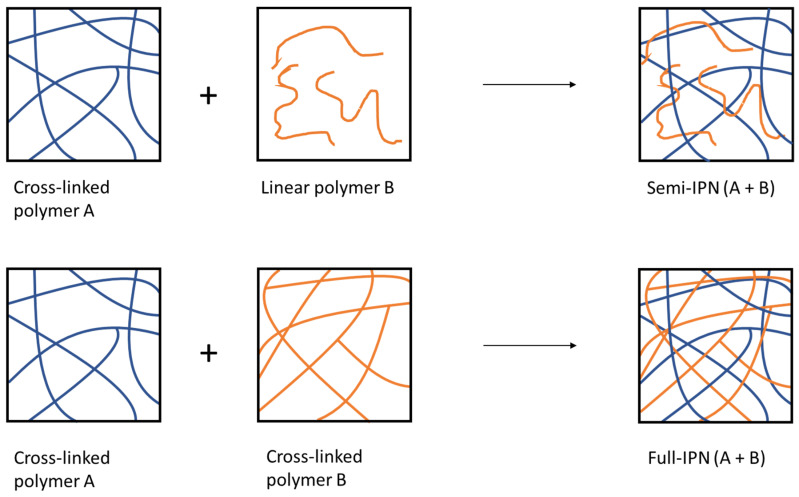
Schematic representation of semi-IPN and full-IPN.

## Data Availability

The data presented in this study are available on request from the corresponding author.

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
