# Peer review of "Optimization of Chitosan Properties with the Aim of a Water Resistant Adhesive Development"

_polymers, 2021, doi:10.3390/polym13224031_

Round 1
Reviewer 1 Report
The title of reviewed manuscript, by Silvestre et al., Optimization of chitosan adhesive properties in the field of bonding, suggests focusing attention on adhesive properties of chitosan. although all the topics discussed are correctly described, their relationship with the adhesive properties is very poorly emphasized. Thus, the reviewed article is another work on the general properties of chitosans, without bringing any scientific novelty. Overall good quality of the text will be good for publication as long as the adjudication properties are highlighted.
Author Response
Thank you for your reviewing. Please find below our answer.The title of reviewed manuscript, by Silvestre et al., Optimization of chitosan adhesive properties in the field of bonding, suggests focusing attention on adhesive properties of chitosan. although all the topics discussed are correctly described, their relationship with the adhesive properties is very poorly emphasized. Thus, the reviewed article is another work on the general properties of chitosans, without bringing any scientific novelty. Overall good quality of the text will be good for publication as long as the adjudication properties are highlighted.
The title of the review has been modified to better correspond to the text content
Reviewer 2 Report
The review paper is relatively well organized. Moreover, the text is well written and there is not any significant weakness which decrease the overall quality of the manuscript.
However, there are some points related to manuscript organization which have to be still optimized:
- Fig. 1: Chemical formulas have to be given in the same size and colour as the others in following Figs.
- There is a duplicity in Figs. 2 and 3 (pages 7-9). The second Figure 2 has to be Figure 4 and the second figure 3 has to be figure 5.
- Fig. 4 is Fig. 6. In the text (line 369) it is correct.
- Fig. 5 is Fig. 7. In the text (line 401) it is correct.
- Fig. 6 is Fig. 8. In the text (line 416) it is correct.
- Fig. 7 is Fig. 9. In the text (line 477) it is correct.
All in all, the review paper is of enough interest. However, a minor revision has to be performed.
Author Response
Thank you for your reviewing. Please find below our answer.
The review paper is relatively well organized. Moreover, the text is well written and there is not any significant weakness which decrease the overall quality of the manuscript.
However, there are some points related to manuscript organization which have to be still optimized:
- Fig. 1: Chemical formulas have to be given in the same size and colour as the others in following Figs.
The figure 1 has been revisited in order to fit with the other figures police
- There is a duplicity in Figs. 2 and 3 (pages 7-9). The second Figure 2 has to be Figure 4 and the second figure 3 has to be figure 5.
- Fig. 4 is Fig. 6. In the text (line 369) it is correct.
- Fig. 5 is Fig. 7. In the text (line 401) it is correct.
- Fig. 6 is Fig. 8. In the text (line 416) it is correct.
- Fig. 7 is Fig. 9. In the text (line 477) it is correct.
All the figures mentioned in the text have been revised
All in all, the review paper is of enough interest. However, a minor revision has to be performed.
Round 2
Reviewer 1 Report
Proposed title is certainly more adequate to the paper content.